# Using geographically weighted regression analysis to assess predictors of home birth hot spots in Ethiopia

**Samuel Hailegebreal**[1]*, **Firehiwot Haile**[2], **Yosef Haile**[2], **Atsedu Endale Simegn**[3], **Ermias Bekele Enyew**[4]

1 School of Public Health Department of Health Informatics, College of Medicine and Health Sciences, Wachemo University, Hosaena, Ethiopia, 2 School of Public Health, College of Medicine and Health Sciences, Arba Minch University, Arba Minch, Ethiopia, 3 Department of Anesthesia, Wachemo University, Hosaena, Ethiopia, 4 Department of Health Informatics, Institute of Public Health, Mettu University, Mettu, Ethiopia

* samuastd@gmail.com

## Abstract

### Background

Annually, 30 million women in Africa become pregnant, with the majority of deliveries taking place at home without the assistance of skilled healthcare personnel. In Ethiopia the proportion of home birth is high with regional disparity. Also limited evidence on spatial regression and deriving predictors. Therefore, this study aimed to assess the predictors of home birth hot spots using geographically weighted regression in Ethiopia.

### Methods

This study used secondary data from the 2019 Ethiopian Mini Demographic and Health Survey. First, Moran's I and Getis-OrdGi* statistics were used to examine the geographic variation in home births. Further, spatial regression was analyzed using ordinary least squares regression and geographically weighted regression to predict hotspot area of home delivery.

### Result

According to this result, Somalia, Afar, and the SNNPR region were shown to be high risk locations for home births. Women from rural residence, women having no-education, poorest wealth index, Muslim religion follower, and women with no-ANC visit were predictors of home delivery hotspot locations.

### Conclusion

The spatial regression revealed women from rural resident, women having no-education, women being in the household with a poorest wealth index, women with Muslim religion follower, and women having no-ANC visit were predictors of home delivery hotspot regions. Therefore, governmental and other stakeholders should remain the effort to decrease home

**Data Availability Statement:** The data we used for this study is available in the DHS program. A letter of approval for the use of the data was secured from the Measure DHS program and the data set

was downloaded from the website (https://dhsprogram.com/data/available-datasets.cfm).

**Funding:** The authors received no specific funding for this work.

**Competing interests:** The authors declare that they have no competing interests.

**Abbreviations:** ANC, Antenatal Care; CSA, Central Statics Agency; DHS, Demographic and Health Survey; EMDHS, Ethiopia Mini Demographic and Health Survey; SNNPR, Southern Nations, Nationalities, and Peoples' Region; WHO, World Health Organization.

childbirth through access to healthcare services especially for rural resident, strengthen the women for antenatal care visits.

## Introduction

Globally, 295,000 women died due to pregnancy and childbirth in 2017, and 86% of all maternal deaths occurred in sub-Saharan Africa and South Asia [1]. For every woman who dies from causes related to pregnancy, 2.9 per 1000 births experience severe complications and potentially fatal condition [2].

According to United Nations interagency estimates the global maternal mortality ratio declined by 38 per cent or an average annual rate of reduction of 2.9 per cent, which is less than half the 6.4 per cent annual rate needed to achieve the sustainable development goal target of 70 maternal deaths per 100,000 live births [3].

Both direct and indirect factors might result in maternal deaths. For instance, about 80% of maternal mortality is caused by severe infection and high blood pressure during pregnancy (preeclampsia and eclampsia) [4, 5]. In Ethiopia, studies discovered that obstructed labor/uterine rupture (36%), hemorrhage (22%), hypertensive disorders of pregnancy (19%) and sepsis/infection (13%) were the main causes of maternal mortality [6].

Neonatal mortality is also higher in African countries at least 300,000 stillbirths occurred from childbirth complications such as obstructed labor. Among babies born alive, another 290,000 die from birth asphyxia which could be preventable through appropriate intervention by skilled health personnel at birth [7]. In Ethiopia neonatal mortality rate was 20.7 deaths per 1000 live births [8]. A studies identified some of the leading causes of neonatal mortality are prematurity (43.9%), early onset of neonatal sepsis (35.1%), low birth weight (33.4%) and birth asphyxia (21.1%) [9].

Annually, 30 million women in Africa becomes pregnant, with the majority of deliveries taking place at home without the assistance of skilled healthcare personnel [10, 11]. In developed countries like the United States more than 98 percent of child birth were taking place at health facility [12] but in sub-Saharan African countries the prevalence of child birth out of health facility was 44% [10]. In Ethiopia the pooled prevalence of child birth at home was 66.7% which is higher than other countries [13]. The proportion of home delivery in Ethiopia varies from place to place it ranges from 80% in Benishangul Gumuz to 28.7% in Jimma Zone [14–18].

The risk of maternal morbidity and mortality is high when a child is born at home without the assistance of a trained health professional [19], Moreover, it has an effect on neonatal outcomes. Current international public health strategy places a strong emphasis on reducing newborn mortality [20]. The setting of delivery is crucial to the provision of reproductive health care. The level of care a woman and child receive is frequently influenced by where they give birth. It has a significant role in determining the varying neonatal mortality risks [21].

Several studies have identified various factors that play a significant role in the observed proportion of home births. This includes mother's educational status, income, residence, distance of health facility, mother's occupation, husband's educational status, ANC visit, media access, maternal age, maternal awareness on danger sign of pregnancy, lack of birth plan or birth preparedness and readiness, home preference as place of child birth and waiting time of greater than during ANC in health facilities were some of the factors that had a significant effect on home birth [14–16, 18, 22–26].

WHO recommends health facility births as a critical intervention to reduce maternal and neonatal mortality since all pregnant women require expert care during birth. Accordingly, those maternal complications that occurred during child birth will be managed timely the mothers' and newborns' lives could be saved [4].

To better understand how home births are distributed geographically, numerous studies have been conducted in Ethiopia [27, 28]. However, No data employing geographically weighted regression analysis and factors influencing home delivery in Ethiopia have been found in the literature, according to our search. Moreover, previous studies did not consider spatial regression analysis which helps us to takes non-stationary variables. In order to tackle the following issues, the current study aimed to employ a spatial analytic technique. First, where are the hotspots (areas with the highest risk) for home delivery situated in Ethiopia? Second, what underlying factors contribute to the spatial variations in home delivery in Ethiopia? Thus, identifying the area-based variability and factors affecting home birth is a crucial first step in creating evidence-based decision-making in prevention and control activities for home deliveries. Therefore, the finding of this study will help policy makers, governmental organizations and nongovernmental organizations to come up with a change in programs and interventions related with reduction of births taking place at home without the help of skilled birth attendants.

## Methods

### Data source

The current study was applied a publically available dataset of EMDHS 2019, and there was no active involvement of patients or members of the public recently; thus, an in-depth examination of data was undertaken without revealing uniqueness in this study. EMDHS 2019 Mini Demographic and Health of its kind in the country performed by the Ethiopian Public Health Institute (EPHI) at the request of the Ministry of Health (MoH). The country is divided into nine regions (Tigray, Amhara, Oromia, Benishangul-Gumuz, Southern Nations, Nationalities and Peoples' Region (SNNP/R), Gambella, Harari), Afar and Somali), and two city administrations (Addis Ababa and Dire Dawa). In 2019 EMDHS, the sample was stratified and selected in two stages. In the first stage, a total of 305 EAs (93 in urban areas and 212 in rural areas) were selected with probability proportional to EA size (based on the 2019 PHC frame) and with independent selection in each sampling stratum. In the second stage, lists of households served as a sampling frame for the selection of households. We retrieved the data from the DHS website (www.dhsprogram.com) after we allowed it by the measure program. A total of 5,527 weighted samples of reproductive-age women who gave birth were included in this study. The detailed sampling procedure has been presented in the full 2019 EMDHS report [29].

### Study variables

**Outcome variable.** We used the outcome variable "place of delivery" and coded as home delivery (when the birth took place at home) and institutional delivery (when the birth took at the hospital, health center, or health post). Finally, a continuous variable called the weighted proportion of home delivery per cluster was employed for spatial analysis, including spatial regression analysis.

**Explanatory variables.** Maternal education, maternal age, religion, sex of household head, marital status, wealth index, ANC visit, birth order, parity, region, and place of residence were considered independent variables.

## Data management and analysis

The data were cleaned by STATA version 14.1 software and Microsoft excel. Whereas the spatial analysis was executed using ArcGIS 10.7. Before conducting spatial analysis, the weighted proportions of home birth (outcome variable) and independent variables performed in STATA and were exported to ArcGIS 10.7.

## Spatial analysis

The global Moran's I statistic was computed to investigation for the presence of spatial auto-correlations. Spatial autocorrelation (Global Moran's I) was done to assess whether the spatial distribution of home delivery was dispersed, clustered, or randomly distributed in Ethiopia [30]. Global Moran's I is spatial statistics used to measure spatial autocorrelation though whole data set to yield single out put ranges from– 1 to 1. Moran's value closed to -1 shows home birth is dispersed, whereas Moran's value close to + 1 indicates home birth is clustered, and if Moran's I value 0 revealed home birth is randomly distributed. The Getis-OrdGi* statistics were used to investigate how spatial autocorrelation varied across the study area by determining the GI* statistic for every area. The statistically significant difference of clustering is determined by the Z-score, and the significance is determined by the p-value. Statistical output with high GI* indicates "hotspot" (high risk) whereas low GI* means a "cold spot"(low risk) of home delivery [31].

## Spatial regression

Spatial regression has both local and global analysis techniques [32]. To ensure the heterogeneity of coefficients across each enumeration area, we handled global geographical regression models first, followed by local geographical analysis [30].

After that, we used exploratory regression along with the appropriate tests to verify the assumptions. The Jarque-Bera test was used to verify the residuals' normality assumption. As residuals are not spatially auto-correlated, confirming the koenker Bp test was done to check if the model under gone for geographically weighted regression or not. Furthermore, to rule out redundancy among independent variable, multicollinearity (VIF<10) was checked.

Geographically weighted regression was executed using ArcGIS 10.7 software. We had also checked the six checks which recommended for a model undergone for spatial regression [33–35]. These coefficients fulfill the previously mentioned two requirements: they have the anticipated sign, there is no overlap in the model's explanatory variables, the coefficients are statistically significant, and they have strong adjusted R2 values. Based on their coefficients, variables that have a p-value under 0.05 are chosen and discussed. Finally, best-fitted model for the data was determined by having the lowest AICc score and a higher adjusted R-squared value [36].

**Ethical consideration.** Ethical clearance for the original EDHS was approved by the Ethiopian Public Health Institute Review Board, Ethiopian Health and Nutrition Research Institute (EHNRI) Review Board, the National Research Ethics Review Committee (NRERC) at the Federal Democratic Republic of Ethiopia Ministry of Science and Technology, the ICF Macro Institutional Review Board, and the Centers for Disease Control and Prevention (CDC). The EMDHS 2019 publications state that each respondent gave written permission to participate. All procedures were followed in accordance with the Helsinki declarations. The data set did not contain any information that was shared with a third party. The research is not an experiment. You can find out more about the DHS's ethical principles and how it uses data at: (http://goo.gl/ny8T6X).

## Result

**Table 1.** Shows that the weighted proportion of home delivery based on place of residence and geographical regions. Among women who delivered at home, 60.0% resided in a rural part of Ethiopia. Likewise, the majority of women who experienced home birth lived in Somali (77%), Afar (72%), Oromia (59%), and followed by SNNPR (52%) region.

### Spatial autocorrelation and significant hot spots of home birth

The spatial autocorrelation analysis revealed that significant spatial variation of home delivery across the country with a global Moran's I value of 0.482 (p-value < 0.001) (**Fig 1**). The statistically significant hotspot areas of home delivery were identified in the Somali, Afar, SNNPR, and parts of Amhara region. While substantial cold spot areas were detected in Benishangul, central Oromia, Addis Ababa, Dire Dawa, and Harari (**Fig 2**).

### Ordinary least square analysis result

The OLS model revealed spatially vary risk factors that affect home delivery in Ethiopia. This model (Adjusted R2 = 0.76) explained around 76% of variability in home birth. The koenker Bp test was found to be statistically significant in this investigation, therefore performing geographically weighted regression is recommended. Because the Jarque-Bera statistic was non-significant (p > 0.10), the model residuals were assumed to be normally distributed. Besides, The Joint Wald statistic was statistically significant (p<0.01), shows that the overall model was significant (**Table 2**). As a result, we run geographically weighted regression and obtained the local coefficients for every explanatory variable. The proportion of women from rural resident, the proportion of women who not attained formal education, the proportion of women in the poorest wealth index, the proportion of women Muslim religion follower, and the proportion of women with no-ANC visit were predictors of home delivery hotspot locations (**Table 2**).

A unit increase for respondent's from rural resident, no-education, poorest wealth quartile, no ANC visit increases home birth by 0.15428, 0.19755, 0.20873, and 0.75384 times

**Table 1. Weighted proportion of home birth by place of residence and regions.**

| Variables | Place of delivery | | |
|---|---|---|---|
| | Institutional N (%) | Home N (%) | Total N (%) |
| **Place of residence** | | | |
| Urban | 962(70) | 405(30) | 1367(24.73) |
| Rural | 1665(40) | 2495(60) | 4160(75.27 |
| **Region** | | | |
| Tigray | 269(72) | 103(28) | 371(6.71) |
| Afar | 24(28) | 62 (72) | 86(1.56) |
| Amhara | 569(54) | 481(46) | 1050(18.99) |
| Oromia | 906(41) | 1305(59) | 2211(40.00) |
| Somali | 95(23) | 313(77) | 409(7.40) |
| Benishangul | 43(64) | 24(36) | 67(1.21) |
| SNNPR | 525(48) | 581(52) | 1106(20.01) |
| Gambela | 17(70) | 7(30) | 25(0.45) |
| Harari | 10(64) | 6(36) | 16(0.29) |
| Addis adaba | 148(95) | 8(5) | 156(2.82) |
| Dire dawa | 21(69) | 9(31) | 30(0.54) |

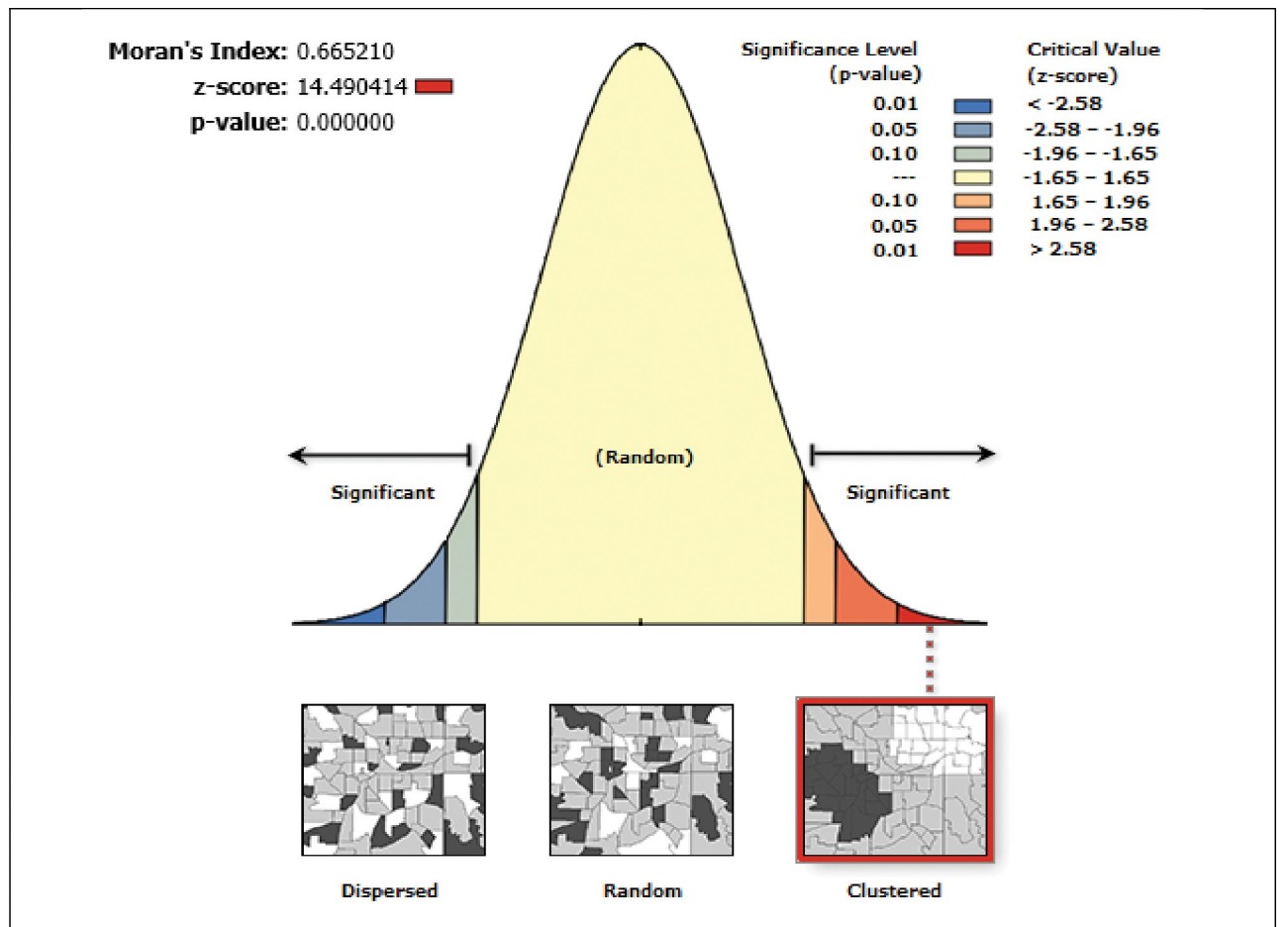

**Fig 1. The global spatial autocorrelation analysis of home birth.**

respectively. On other hand, a unit increase Muslim religion decreases home birth by 0.14867 times (**Table 2**).

## Geographically weighted regression analysis

The GWR analysis showed that the global model had been improved significantly. The AICc value declined from -213.18 in the OLS model to -229.86 the GWR model. In addition, the adjusted R2 (0.76) obtained from OLS increased to adjusted R2 (0.79), implying that GWR improved the model's ability to predict home birth. Overall, according to this study GWR analysis performed better than the model generated using OLS (**Table 3**).

In this study GWR showed that the explanatory variables were both strong and weak predictors of home. As the proportion of women from rural resident increased, the percentage of home birth increased in the entire Gambella, western Oromia, Harari Somali and parts of SNNP regions (**Fig 3**).

Women who had not attended formal education had a strong positive relationship with home birth. As the proportion of women who had not attended formal education increased, the existence of home birth in Tigray, Amhara, Afar, and northern Somali increased (**Fig 4**). As shown in (**Fig 5**) home birth higher coefficients of poorest wealth index were detected in Tigray, Amhara, Benishangul, parts of Oromia and a few parts of Afar region.

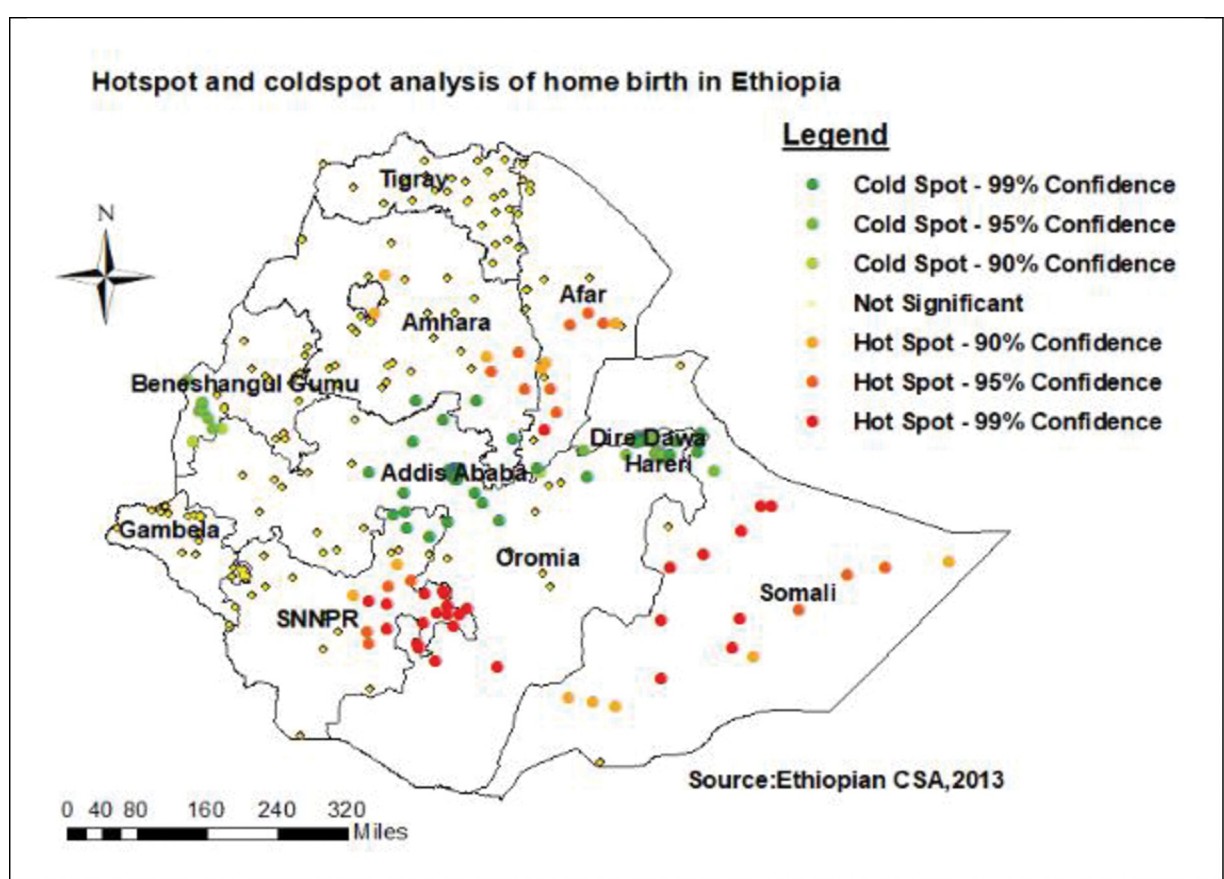

**Fig 2. Hot spots and cold spot analysis of home birth in Ethiopia.**

**Table 2. Summary of ordinary least squares result.**

| Variable | Coefficient | Standard error (SE) | t -Statistic | Probability | Robust standard error | Robust t-statistics | Robust probability | VIF |
|---|---|---|---|---|---|---|---|---|
| Intercept | -0.075 | 0.023 | -3.26 | < 0.01 | 0.0185 | -4.06 | < 0.01 | ------- |
| proportion of women from rural resident | 0.15428 | 0.025 | 6.08 | < 0.01 | 0.023 | 6.66 | < 0.01 | 1.46 |
| proportion of women who not attained formal education | 0.19755 | 0.044 | 4.46 | < 0.01 | 0.046 | 4.28 | < 0.01 | 1.97 |
| proportion of women in the poorest wealth index | 0.20873 | 0.041 | 5.00 | < 0.01 | 0.043 | 4.80 | < 0.01 | 2.12 |
| proportion of women Muslim religion follower | -0.14867 | 0.024 | -5.98 | < 0.05 | 0.025 | -5.89 | < 0.05 | 1.30 |
| Proportion of women no ANC visit | 0.75384 | 0.058 | 12.86 | < 0.01 | 0.064 | 11.67 | < 0.01 | 2.37 |
| **OLS diagnosis** | | | | | | | | |
| Number of Observations: | 305 | Akaike's Information Criterion (AICc) | | | | | | -213.178 |
| Multiple R-Squared | 0.77 | Adjusted R-Squared | | | | | | 0.768 |
| Joint F-Statistic | 203.06 | Prob(>F), (5,299) degrees of freedom | | | | | | < 0.01 |
| Joint Wald Statistic: | 1746.57 | Prob(>chi-squared), (5) degrees of freedom | | | | | | < 0.01 |
| Koenker (BP) Statistic | 26.41 | Prob(>chi-squared), (5) degrees of freedom | | | | | | < 0.01 |
| Jarque-Bera Statistic | 3.31 | Prob(>chi-squared), (2) degrees of freedom | | | | | | 0.191 |

**Table 3. Geographic weighted regression (GWR) model for the home birth in Ethiopia.**

| Explanatory variables | Rural resident, No-education, Poorest wealth, Muslim religion, No ANC visit |
|---|---|
| Residual squares | 6.93 |
| Effective number | 35.10 |
| Sigma | 0.16 |
| Akaike's Information Criterion (AICc) | -229.86 |
| Multiple R-Squared | 0.81 |
| Adjusted R-square | 0.79 |

Regarding to Muslim religion, women with Muslim religion followers had a strong and negative relationship with home delivery in the Tigray, and the central part of Afar (**Fig 6**). Furthermore, women without ANC visit had a strong and positive relationship with childbirth at home in the eastern part of SNNPR, western Oromia, and some parts of Somali regions (**Fig 7**).

## Discussion

This study aimed to determine the geographical variations of home birth and its predictors in Ethiopia using geographical weighted regression models. In this finding highly clustering home birth observed in the Somali, Afar, SNNPR, and parts of Amhara region. Previous studies revealed that significant geographical variations in home delivery [27, 37, 38]. The potential geographical disparity of home delivery across region could be due to regional inaccessible

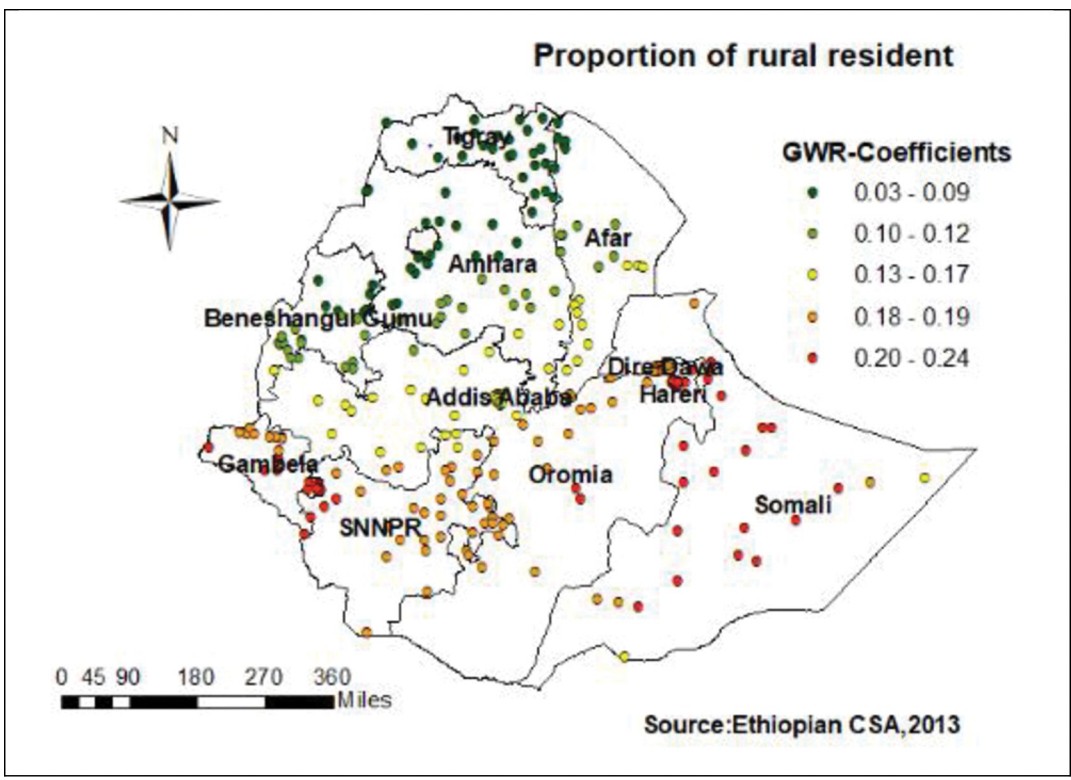

**Fig 3. Women from rural resident GWR coefficients for predicting home birth in Ethiopia.**

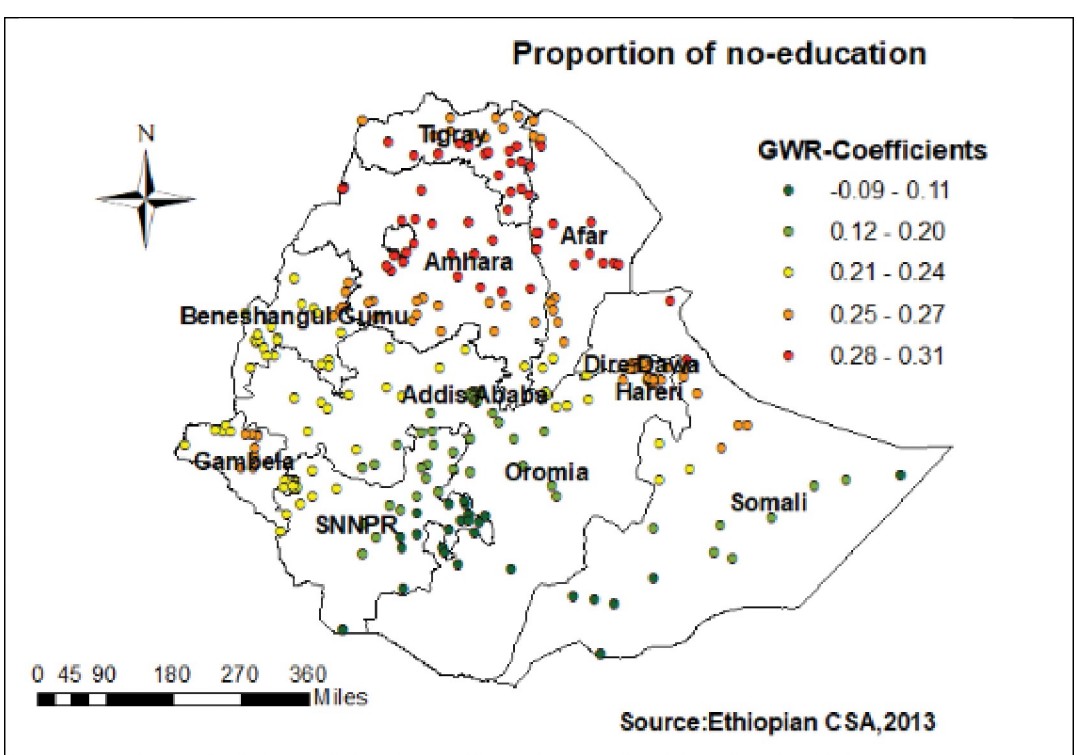

**Fig 4. Women with no education GWR coefficients for predicting home birth in Ethiopia.**

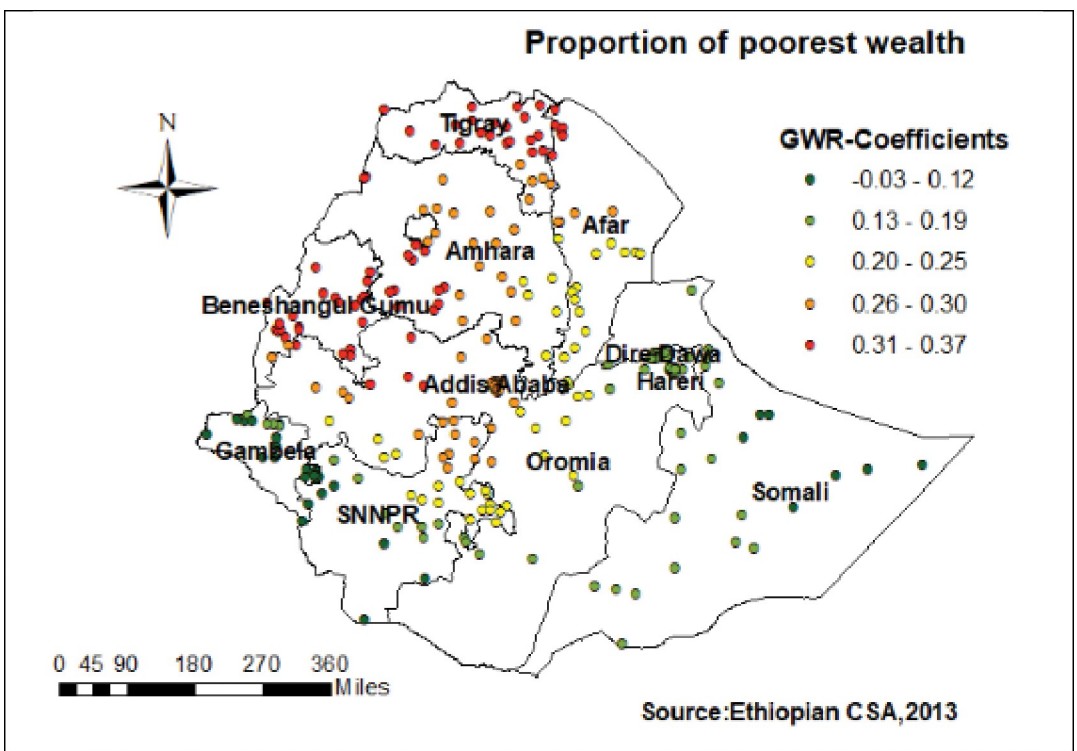

**Fig 5. Women from poorest household GWR coefficients for predicting home birth in Ethiopia.**

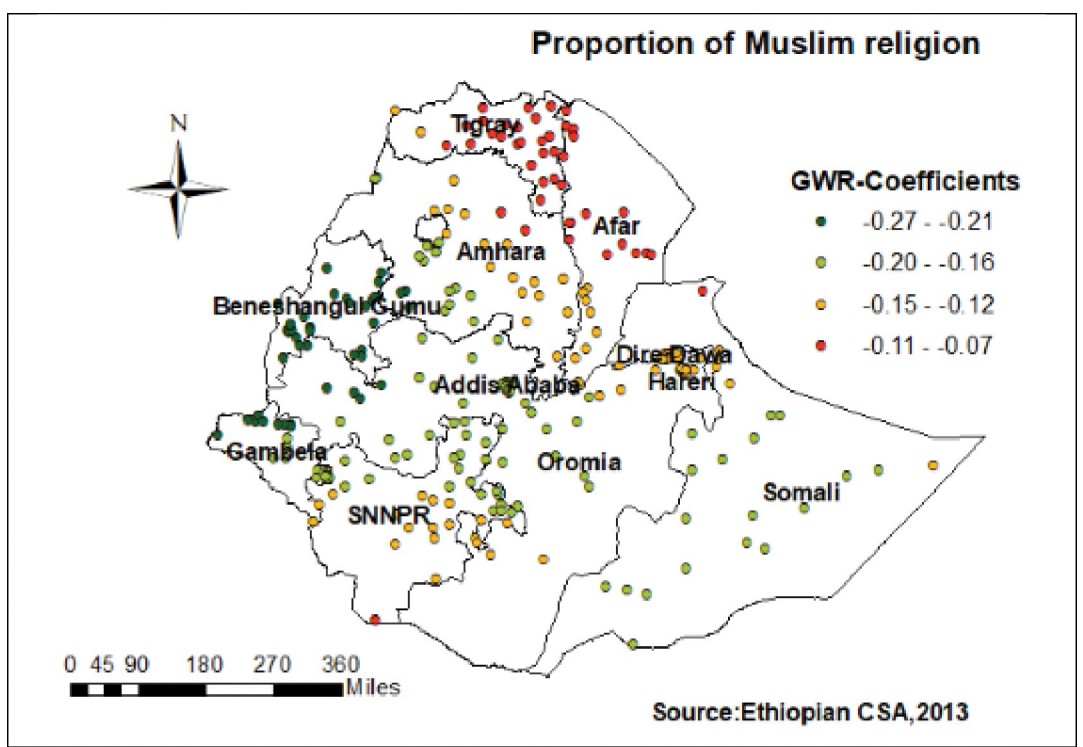

**Fig 6. Women with Muslim religion GWR coefficients for predicting home birth in Ethiopia.**

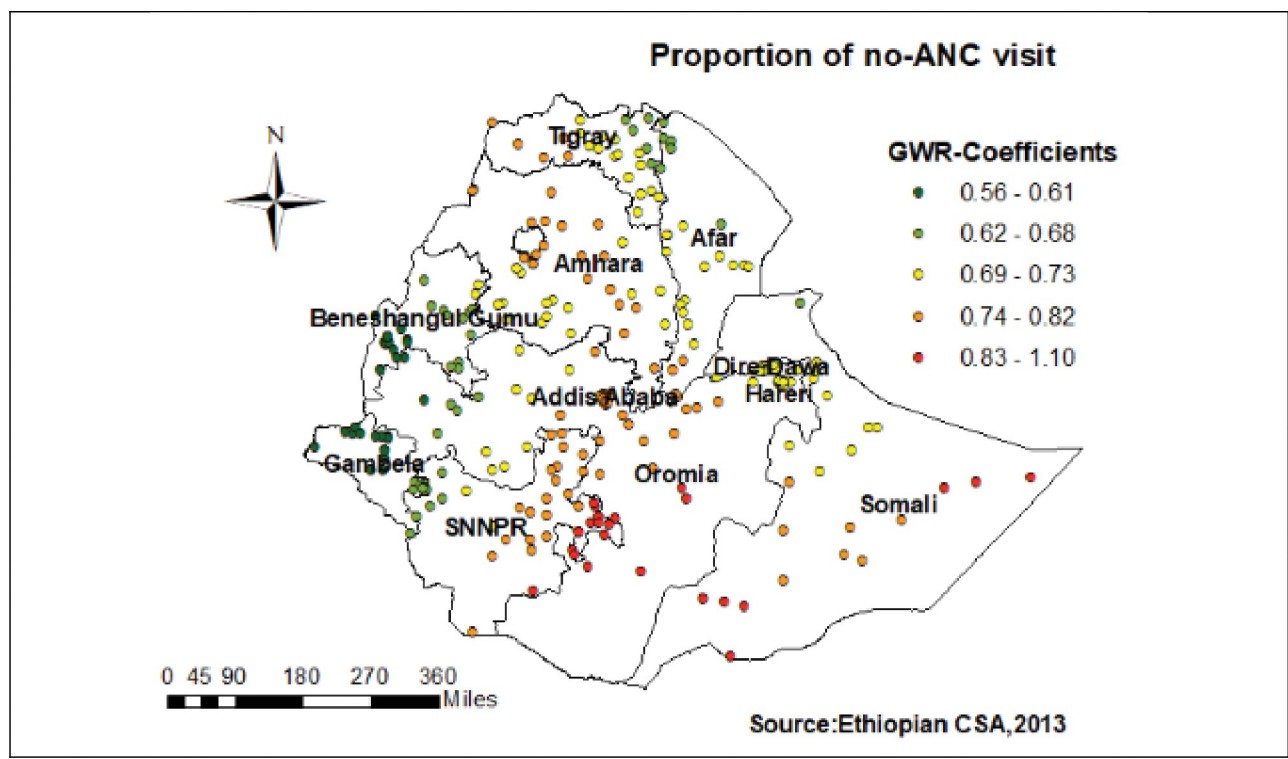

**Fig 7. Women with no ANC visit GWR coefficients for predicting home birth in Ethiopia.**

health facilities. In addition, this might socio-demographic, economic and cultural variation, between women in crosswise region in the country. Besides, women from boundary areas might have inadequate access to information concerning parental services [39].

In the GWR regression analysis, statistically significant predictor variables were identified that had strong positive and negative relationship with home delivery. The proportion of women from rural residence, the proportion of women who not attained formal education, the proportion of women in the poorest wealth index, the proportion of women with Muslim religion follower, and the proportion of women with no-ANC visit were predictors of home delivery hotspot locations.

The GWR coefficients of home delivery for those women who reside rural residence were varied between 0.03 and 0.24 from region to region (**Fig 3**). The proportion of rural resident strongly predicts the experience of home delivery in the entire Gambella, western Oromia, Harari Somali and parts of SNNP region. This could be associated to rural residence lack access to health information and facilities, as well as their distance from a health facility, which increased the possibility of home delivery [40, 41].

The study also highlighted that women who had not attended formal education had a strong positive relationship with home delivery. The GWR coefficients of this proportion of no-education varied from 0.09 to 0.31 across geographic region. It strongly predicts the existence of home birth in Tigray, Amhara, Afar, and northern Somali. The fact that educated women would distinguish the advantage of institutional delivery and the threat of birth at home through different media source like reading magazines, broadcasting, mass media, and social media [42].

Similarly, the proportion of women in the poorest wealth index had positive relationships to home delivery. In this finding higher GWR coefficients of poorest wealth index were detected in Tigray, Amhara, Benishangul, Oromia and a few parts Afar region. This might be due to transportation costs for women in labor and the families who accompany them to a health facility may be a challenge in poorest wealth index families and may impact pregnant women to choose for home childbirth [43]. There are also costs desired to prepare birth items demanded by health care providers [44]. However, Muslim religion followers had a negative relationship with hot spots of home delivery in Tigray, and the central part of Afar. The possible reason might be Muslim women who believe that their naked bodies can only be seen by their husbands may select home births by traditional birth attendant than health facility delivery by skilled personnel [20, 45].

Moreover, women without ANC visit had a strong and positive relationship with childbirth at home in the eastern part of SNNPR, western Oromia, and some parts of Somali regions.

The possible explanation could be during ANC visit they get the chance of taking additional knowledge on the significance of facility delivery over than home delivery [46]. Besides, makes good opportunity on increasing advising the women from skilled personnel which have excessive aptitude to decrease home child birth [13].

The strength of this study was the use of data from a nationally representative a big dataset, which results in sufficient statistical power. Modeling spatial regression relationships using GWR was also the strength of this study. As a limitation, for the privacy issue, the geographic locations of enumeration areas were displaced up to 2 km in urban areas, 5 km for rural areas, which may have an impact on the spatial regression. The other limitation was that some important possible factors that could affect the practice like mother's occupation, and father's characteristics were not included due to incompleteness of information.

## Conclusions

Significant hotspot areas of home delivery were identified in the Somali, Afar, SNNPR regions. The local spatial regression (GWR) revealed women from rural residence, women having no-

education, women in the poorest wealth index, women with Muslim religion followers, and women with no-ANC visit were predictors of home childbirth hotspot area. Therefore, governmental and other stakeholders should remain the effort to decrease home childbirth through access to healthcare services especially for rural resident, strengthen the women for antenatal care visits.

## Acknowledgments

We would like to express our deepest thankfulness to Measure DHS, for providing the data for the study.

## Author Contributions

**Conceptualization:** Samuel Hailegebreal, Ermias Bekele Enyew.

**Data curation:** Yosef Haile.

**Formal analysis:** Samuel Hailegebreal, Firehiwot Haile.

**Investigation:** Yosef Haile, Atsedu Endale Simegn.

**Methodology:** Samuel Hailegebreal.

**Resources:** Firehiwot Haile, Ermias Bekele Enyew.

**Software:** Samuel Hailegebreal.

**Supervision:** Atsedu Endale Simegn.

**Writing – original draft:** Samuel Hailegebreal.

**Writing – review & editing:** Firehiwot Haile, Yosef Haile, Atsedu Endale Simegn, Ermias Bekele Enyew.

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
