## [Decision Letter · Decision Letter 0]

4 Nov 2022

PONE-D-22-09551Using geographically weighted regression analysis to assess predictors of home birth hot spots in EthiopiaPLOS ONE

Dear Dr. Hailegebreal,

Thank you for submitting your manuscript to PLOS ONE. After careful consideration, we feel that it has merit but does not fully meet PLOS ONE’s publication criteria as it currently stands. Therefore, we invite you to submit a revised version of the manuscript that addresses the points raised during the review process.

We look forward to receiving your revised manuscript.

Kind regards,

Hermano Alexandre Lima Rocha

Academic Editor

PLOS ONE

Journal Requirements:

“no computing interest”

7. We note that Figures 2-7 in your submission contain [map/satellite] images which may be copyrighted. All PLOS content is published under the Creative Commons Attribution License (CC BY 4.0), which means that the manuscript, images, and Supporting Information files will be freely available online, and any third party is permitted to access, download, copy, distribute, and use these materials in any way, even commercially, with proper attribution. For these reasons, we cannot publish previously copyrighted maps or satellite images created using proprietary data, such as Google software (Google Maps, Street View, and Earth). For more information, see our copyright guidelines: http://journals.plos.org/plosone/s/licenses-and-copyright.

a. You may seek permission from the original copyright holder of Figures 2-7 to publish the content specifically under the CC BY 4.0 license. 

Additional Editor Comments (if provided):

Dear authors, As you can imagine, many manuscripts need adjustments after the first round of review. Please read carefully the points highlighted by the experts who reviewed your work and send us a new version with the requested adjustments. Best wishes

Reviewers' comments:

Reviewer's Responses to Questions

**Comments to the Author**

1. Is the manuscript technically sound, and do the data support the conclusions?

Reviewer #1: No

Reviewer #2: Yes

2. Has the statistical analysis been performed appropriately and rigorously? 

Reviewer #1: No

Reviewer #2: Yes

3. Have the authors made all data underlying the findings in their manuscript fully available?

Reviewer #1: Yes

Reviewer #2: Yes

4. Is the manuscript presented in an intelligible fashion and written in standard English?

Reviewer #1: Yes

Reviewer #2: Yes

5. Review Comments to the Author

Reviewer #1: Major general concern

From the spatial analysis, it is obvious the analysis was done at the regional level; however, the authors did not include any regional level variables in the study. But instead, an aggregation of the individual-level variables were used. How did the authors control the heterogeneity of the various enumeration areas making up a region? Also, the authors must read the concept of ecological fallacy and indicate how they corrected that in their study? Because they used individual-level aggregated data to represent the regions.

Background

The background of the manuscript is well written and clearly articulates the burden of home birth. However, there is not enough scientific justification besides the authors indicating that spatial regression considers non-stationary variables. Furthermore, despite indicating that the manuscript accounts for non-stationary variables, the authors failed to state what non-stationary variables were included in their study and how different they are from previous studies.

Methods

the outcome variable considered in the study is a dichotomised variable based on its description; the authors failed to indicate how they performed a GWR, which takes a continuous dependent variable and continuous independent variables.

The list of explanatory variables appears limited since the variables influencing home birth go beyond the individual and household variables. In addition, the authors failed to account for variables such as distance to facility, health insurance and cost related to health care utilisation.

The authors didn’t describe how the weighted proportion was estimated and at what level they measured the weighted proportion of home birth.

Results

The results presented in Table 2: Summary of ordinary least squares result does not account for all the explanatory variables stated in the methods section. The authors did not provide any justification after stating that all variables were considered.

Reviewer #2: See attached word document.

6. PLOS authors have the option to publish the peer review history of their article (what does this mean?). If published, this will include your full peer review and any attached files.

Reviewer #1: No

Reviewer #2: No

---

## [Author Response · Author response to Decision Letter 0]

9 Dec 2022

To: PLOS ONE 

From: Samuel Hailegebreal 

Subject: A letter Accompanying Revision in Response to Editors and Reviewer Comments

Dear Editors 

The authors would like to thank the editorial team and team of reviewers for constructive and valuable comments. The authors are very happy to submit the revised version of the manuscript entitled “Using geographically weighted regression analysis to assess predictors of home birth hot spots in Ethiopia” for its publication in your Journal. The comments of the editors and the reviewers were highly insightful and enabled us to greatly improve the quality of our manuscript. In this revised manuscript we made substantial changes to address your concerns in a point-by-point response. We are very keen to incorporate further comments, if any, for the betterment of the final manuscript.

Point by Point Response to - Editor Comments 

1. We note that Figures 2-7 in your submission contain [map/satellite] images which may be copyrighted. All PLOS content is published under the Creative Commons Attribution License (CC BY 4.0), which means that the manuscript, images, and Supporting Information files will be freely available online, and any third party is permitted to access, download, copy, distribute, and use these materials in any way, even commercially, with proper attribution. For these reasons, we cannot publish previously copyrighted maps or satellite images created using proprietary data, such as Google software (Google Maps, Street View, and Earth). For more information, see our copyright guidelines: http://journals.plos.org/plosone/s/licenses-and-copyright.

Authors’ response: Thank you editor for the concern. The map is not copyrighted rather we have done using ArcGIS and SaTScan software based on the shapefile of Ethiopia received from Ethiopian Central Statistical Agency (CSA) by explaining the purpose of the study and GPS data (longitude and latitude) from measure DHS program by explaining the objective of the study through online requesting and allow us to access the shapefile and GPS data. Now we cite the source of the shapefile since it is needed to explore the spatial distribution of home delivery. Therefore, the maps presented in our study are not copyrighted rather it was our spatial analysis result.

Point by Point Response to Reviewers 

Reviewer #1

1) From the spatial analysis, it is obvious the analysis was done at the regional level; however, the authors did not include any regional level variables in the study. But instead, an aggregation of the individual-level variables was used. How did the authors control the heterogeneity of the various enumeration areas making up a region? Also, the authors must read the concept of ecological fallacy and indicate how they corrected that in their study? Because they used individual-level aggregated data to represent the regions.

Response: - we appreciate your thoughtful observation. Since we haven't yet combined individual level variables with community level variables in this study, there isn't a scenario for which we would have to be concerned about this type of fallacy. Instead, we have used regional variables (community level) like place of residence and region (see it in the method section).

2) The background of the manuscript is well written and clearly articulates the burden of home birth. However, there is not enough scientific justification besides the authors indicating that spatial regression considers non-stationary variables. Furthermore, despite indicating that the manuscript accounts for non-stationary variables, the authors failed to state what non-stationary variables were included in their study and how different they are from previous studies

Response: - Thank you for your valuable comment and we updated in the revised version “To better understand how home births are distributed geographically, numerous studies have been conducted. However, the quantity of geographical research on home birth is constrained by a lack of modeling of the spatial relationships between the reported clusters of home deliveries and their determinants. Moreover, previous studies did not consider spatial regression analysis which helps us to takes non-stationary variables. In order to tackle the following issues, the current study aimed to employ a spatial analytic technique. First, where are the hotspots (areas with the highest risk) for home delivery situated in Ethiopia? Second, what underlying factors contribute to the spatial variations in home delivery in Ethiopia? Thus, identifying the area-based variability and factors affecting home birth is a crucial first step in creating evidence-based decision-making in prevention and control activities for home deliveries”

3) The outcome variable considered in the study is a dichotomised variable based on its description; the authors failed to indicate how they performed a GWR, which takes a continuous dependent variable and continuous independent variables.

The list of explanatory variables appears limited since the variables influencing home birth go beyond the individual and household variables. In addition, the authors failed to account for variables such as distance to facility, health insurance and cost related to health care utilisation. The authors didn’t describe how the weighted proportion was estimated and at what level they measured the weighted proportion of home birth.

Response: - Thank you for the valuable comment and updated in the revised version. Finally, a continuous variable called the weighted proportion of home delivery per cluster was employed for spatial analysis, including spatial regression analysis. Due to the fact that the variables (such as distance to facility, health insurance and cost related to health care utilisation) you indicate were not gathered in MDHS Ethiopia 2019, we regarded this as a restriction.

4) The results presented in Table 2: Summary of ordinary least squares result does not account for all the explanatory variables stated in the methods section. The authors did not provide any justification after stating that all variables were considered.

Response: - thank you! Because the variables listed in the technique section failed to support the aforementioned assumption, only those included in the OLS results were predictive of home birth in our study. You can find the justification (see it in ordinary least square result section)

Reviewer #2

1. Consider keeping information focused to African countries for this section, unless there is no information about the topic in that region (e.g., is line 98 needed about USA?)

Response: - Thank you for your valuable comment we accepted and corrected in the revised version 

2. Line 106: there is a wealth of literature on socio-cultural factors that influence home birth in sub-Saharan Africa. A few should be included here.

Response: - We author thanks for you valuable comment we accept and we have add few more in the revised version 

3. Lines 111-116: consider stating the primary and secondary aims of your study here.

Response: Thank you we accepted and corrected in the revised version “In order to tackle the following issues, the current study aimed to employ a spatial analytic technique. First, where are the hotspots (areas with the highest risk) for home delivery situated in Ethiopia? Second, what underlying factors contribute to the spatial variations in home delivery in Ethiopia? Thus, identifying the area-based variability and factors affecting home birth is a crucial first step in creating evidence-based decision-making in prevention and control activities for home deliveries.”

4. Line 131=132: this sentence is confusing, I’m not sure what it means- consider omitting it

Response: - We accept the comment and corrected in the revised version 

5. Spatial regression section: Much of this is not necessary and could be included in an appendix. Rather than explain the method itself, only keep in the text what you did in your analysis (do not need to explain the method itself).

Spatial regression

“Spatial regression has both local and global analysis techniques (30). To ensure the heterogeneity of coefficients across each enumeration area, we handled global geographical regression models first, followed by local geographical analysis (28). After that, we used exploratory regression along with the appropriate tests to verify the assumptions. The Jarque-Bera test was used to verify the residuals' normality assumption. As residuals are not spatially auto-correlated, confirming the koenker Bp test was done to check if the model under gone for geographically weighted regression or not. Furthermore, to rule out redundancy among independent variable, multicollinearity (VIF<10) was checked. Geographically weighted regression was executed using ArcGIS 10.7 software. We had also checked the six checks which recommended for a model undergone for spatial regression (31, 32, 33). These coefficients fulfill the previously mentioned two requirements: they have the anticipated sign, there is no overlap in the model's explanatory variables, the coefficients are statistically significant, and they have strong adjusted R2 values. Based on their coefficients, variables that have a p-value under 0.05 are chosen and discussed. Finally, best-fitted model for the data was determined by having the lowest AICc score and a higher adjusted R-squared value (34)”

6. Lines 202-210: If this section begins your results, then have a heading that says Results above it to orient the reader

Response: - Thank you for your comment we accept and corrected 

7. Please omit explanatory sentences about the methods use and stick to just reporting the results here. Further for lines 211-214, need to include the results in the text to match Table 2 for each socio-demographic that is listed.

Response: - Thank you for the comment we omitted explanatory sentence and updated the table accordingly.

8. Table 2: do not need to include OLS diagnosis- this could be included in appendices but not necessary

Response: - We appreciate your concern. The reason we provide OLS diagnostic here is so that academics may determine whether or not the assumption feet is accurate.

---

## [Decision Letter · Decision Letter 1]

14 Mar 2023

PONE-D-22-09551R1Using geographically weighted regression analysis to assess predictors of home birth hot spots in EthiopiaPLOS ONE

Dear Dr. Gele,

Thank you for submitting your manuscript to PLOS ONE. After careful consideration, we feel that it has merit but does not fully meet PLOS ONE’s publication criteria as it currently stands. Therefore, we invite you to submit a revised version of the manuscript that addresses the points raised during the review process.

We look forward to receiving your revised manuscript.

Kind regards,

Hermano Alexandre Lima Rocha

Academic Editor

PLOS ONE

Journal Requirements:

Reviewers' comments:

Reviewer's Responses to Questions

**Comments to the Author**

1. If the authors have adequately addressed your comments raised in a previous round of review and you feel that this manuscript is now acceptable for publication, you may indicate that here to bypass the “Comments to the Author” section, enter your conflict of interest statement in the “Confidential to Editor” section, and submit your "Accept" recommendation.

Reviewer #1: All comments have been addressed

2. Is the manuscript technically sound, and do the data support the conclusions?

Reviewer #1: Partly

3. Has the statistical analysis been performed appropriately and rigorously? 

Reviewer #1: Yes

4. Have the authors made all data underlying the findings in their manuscript fully available?

Reviewer #1: Yes

5. Is the manuscript presented in an intelligible fashion and written in standard English?

Reviewer #1: Yes

6. Review Comments to the Author

Reviewer #1: 1. Line 102 -108 have different font types

2. There are inherent grammatical errors

3. the authors stated " a lack of modeling of the spatial relationships" this statement indicated that they didn't do an extensive literature search.

4. The authors failed to mention the non-stationary variables which are being included in the study

7. PLOS authors have the option to publish the peer review history of their article (what does this mean?). If published, this will include your full peer review and any attached files.

Reviewer #1: No

---

## [Author Response · Author response to Decision Letter 1]

16 Mar 2023

To: PLOS ONE 

From: Samuel Hailegebreal 

Subject: A letter Accompanying Revision in Response to Editors and Reviewer Comments

Dear Editors 

The authors would like to thank the editorial team and team of reviewers for constructive and valuable comments. The authors are very happy to submit the revised version of the manuscript entitled “Using geographically weighted regression analysis to assess predictors of home birth hot spots in Ethiopia” for its publication in your Journal. The comments of the editors and the reviewers were highly insightful and enabled us to greatly improve the quality of our manuscript. In this revised manuscript we made substantial changes to address your concerns in a point-by-point response. We are very keen to incorporate further comments, if any, for the betterment of the final manuscript.

Point by Point Response to - Editor Comments 

Response: we change reference 6,7,10 and also we added 29 and 30 in revised manuscript 

Point by Point Response to Reviewers 

Reviewer #1

1. Line 102 -108 have different font types

Response: we corrected in the revised version 

2. There are inherent grammatical errors

Response: we edited carefully spelling and grammatical error and also we used the link provided by reviewer Pre-publication Support Service 

3. The authors stated “a lack of modeling of the spatial relationships" this statement indicated that they didn't do an extensive literature search.

Response: I appreciate your comment. There are few studies on spatial distribution home delivery in Ethiopia. Yet, according to our search, no study on the place of delivery in Ethiopia has ever used geographically weighted regression.

4. The authors failed to mention the non-stationary variables which are being included in the study

Response: I appreciate your input. Since the EDHS 2019 is interim significant characteristics were not included, this was highlighted in the limitation section.

---

## [Editor Report · Decision Letter 2]

23 May 2023

Using geographically weighted regression analysis to assess predictors of home birth hot spots in Ethiopia

PONE-D-22-09551R2

Dear Dr. Hailegebreal,

We’re pleased to inform you that your manuscript has been judged scientifically suitable for publication and will be formally accepted for publication once it meets all outstanding technical requirements.

Kind regards,

Miquel Vall-llosera Camps

Senior Editor

PLOS ONE
---

## [Editor Report · Acceptance letter]

29 May 2023

PONE-D-22-09551R2 

Using geographically weighted regression analysis to assess predictors of home birth hot spots in Ethiopia 

Dear Dr. Hailegebreal:

I'm pleased to inform you that your manuscript has been deemed suitable for publication in PLOS ONE. Congratulations! Your manuscript is now with our production department. 

Kind regards, 

on behalf of

Dr. Miquel Vall-llosera Camps 

Staff Editor

PLOS ONE